# Memristor-Based Signal Processing for Compressed Sensing

**DOI:** 10.3390/nano13081354

**Published:** 2023-04-13

**Authors:** Rui Wang, Wanlin Zhang, Saisai Wang, Tonglong Zeng, Xiaohua Ma, Hong Wang, Yue Hao

**Affiliations:** 1Key Laboratory of Wide Band Gap Semiconductor Technology, School of Microelectronics, Xidian University, Xi’an 710071, China; 2Key Laboratory of Wide Band Gap Semiconductor Technology, School of Advanced Materials and Nanotechnology, Xidian University, Xi’an 710071, China

**Keywords:** memristor, inherent variation, compressed sensing, compression and encryption, edge computing

## Abstract

With the rapid progress of artificial intelligence, various perception networks were constructed to enable Internet of Things (IoT) applications, thereby imposing formidable challenges to communication bandwidth and information security. Memristors, which exhibit powerful analog computing capabilities, emerged as a promising solution expected to address these challenges by enabling the development of the next-generation high-speed digital compressed sensing (CS) technologies for edge computing. However, the mechanisms and fundamental properties of memristors for achieving CS remain unclear, and the underlying principles for selecting different implementation methods based on various application scenarios have yet to be elucidated. A comprehensive overview of memristor-based CS techniques is currently lacking. In this article, we systematically presented CS requirements on device performance and hardware implementation. The relevant models were analyzed and discussed from the mechanism level to elaborate the memristor CS system scientifically. In addition, the method of deploying CS hardware using the powerful signal processing capabilities and unique performance of memristors was further reviewed. Subsequently, the potential of memristors in all-in-one compression and encryption was anticipated. Finally, existing challenges and future outlooks for memristor-based CS systems were discussed.

## 1. Introduction

With the advent of emerging technologies, particularly in the domain of the Internet of Things (IoT) and artificial intelligence, numerous perceptual networks were devised to achieve seamless human–computer interaction. By 2025, the number of sensing nodes is anticipated to reach 75 billion, and by 2030, it is projected to surpass 125 billion [1,2,3]. With massive amounts of digital information being stored, replicated, processed, and communicated, bandwidth is becoming an increasingly scarce resource. The Shannon–Nyquist sampling theorem stipulates that the sampling rate should be at least twice the bandwidth to uniformly sample a signal without losing any information, which inevitably constrains the sampling speed.

Donoho proposed a revolutionary compression technology called compressed sensing (CS), which triggered widespread interest in signal processing and wireless communication networks [4]. CS employs random matrix coding to achieve simultaneous data compression and acquisition at the interface of the analog and digital domains, enabling sub-Nyquist sampling rates. Specifically, a random Φ-matrix, also referred to as a measurement matrix, is employed as a random matrix to encode a signal for compression. To decode the CS measurement, any receiver must know the actual encoding Φ-matrix used during sampling, and the compressive signal can be accurately recovered by solving convex optimization problems [5]. Compared to the traditional sampling and compression process, this random non-uniform sampling does not need to adhere to Nyquist’s law of sampling, thus enabling low-power and high-efficiency data processing. However, complex sampling control modules and intensive matrix-vector multiplication (MVM) operations can still be cumbersome, limiting scalability and sampling speed. Therefore, there is an urgent need for an efficient hardware solution.

Memristor is one such technology that aims to develop the next generation of high-density and high-speed digital technology through its powerful analog computing capabilities [6,7,8,9,10]. In recent years, considerable progress was made in the development of large-scale and efficient parallel analog computing capabilities based on transition metal oxide memristors. For instance, a Ta/HfO_2_ 128 × 64 1T1R crossbar array was constructed and used for analog signal processing and machine learning [11,12,13,14]; and a 32 × 32 WO_x_-based memristor crossbar array was used for image sparse processing [15]. However, memristors exhibit serious non-ideal effects, such as variability, which are attributed to the uncontrollable ion migration behavior. The non-ideal characteristics associated with these devices hinder systems from achieving high accuracy, as greater variability can lead to signal attenuation in a crossbar array, with selected devices contributing more to signal attenuation than unselected devices [16]. Ongoing research in this area aims to mitigate these “detrimental” effects [17,18,19,20,21]. Prezioso et al. [17] optimized through an exhaustive experimental search for a range of titania compositions and layer thicknesses (from 5 nm to 100 nm). Gao et al. [18] developed oxide-based synapses with 3D vertical structures, incorporating multiple parallel resistive random access memory (RRAM) devices on the same nanopillar to suppress intrinsic variation, resulting in a significant increase in recognition accuracy from 65% to 90%. Interestingly, the non-ideal variability that most existing work on memristor-based systems attempts to mitigate is highly beneficial for CS systems. The most commonly used approach to implement CS using memristors is to leverage the memristor’s random oscillation switching behavior and conductance distribution to realize the measurement matrix Φ. Subsequently, the powerful analog computation capabilities of the memristor are harnessed to perform vector multiplication with the input signal, ultimately achieving signal compression, as depicted in Figure 1. The compressed signal can then be stored in the cloud and, upon request, extracted and reconstructed, making it an attractive technology for numerous edge computing applications, such as cameras, mobile phones, computers, robots, and autonomous driving, as shown in Figure 1.

Currently, with the rapid development of IoT technology and the increasing demand for efficient edge computing, there is an urgent need for an efficient implementation of CS hardware. This review provides a systematic presentation of the requirements for device performance and hardware implementation for CS. The essential electrical properties required for CS implementation are introduced and analyzed from the mechanism level, and relevant models are discussed. In combination with the powerful signal processing capabilities of memristors, it can address the challenge of high-density MVM sampling and accelerate the implementation of CS. From the perspective of the hardware system, the current approaches for implementing memristors-based CS are summarized, including constructing a measurement matrix by pre-programming the array conductance states, controlling sampler sampling through intrinsic variability, and forming a measurement matrix using the conductance state of intrinsic variation. It is noteworthy that memristor-based CS systems are expected to realize the all-in-one compression and encryption engine by embedding the encryption layer in the compression layer to build a zero-cost encryption scheme. Finally, based on the current hardware implementation of memristors, the current challenges and improvement directions are proposed to effectively meet the growing demand for information efficiency and security in rapidly developing data-intensive technologies.

## 2. Investigation of Memristor Inherent Variation

### 2.1. Inherent Variation Metrics Study

Variation, one of the non-ideal characteristics of memristors, refers to the spatial stochastic behaviors observed from device-to-device (D2D) and cycle-to-cycle (C2C) [22,23]. The concept of intrinsic variability in the operation of memristor arrays is not a new feature in unique devices, but rather a study addressing the internal switching phenomenon and its underlying probabilistic nature. This feature was perceived by emerging memory technologies such as phase-change memory (PCM) [24], conductive-bridge memory (CBRAM) [25], and oxide-based memory (OXRAM) [19]. Despite the aforementioned technical differences, the commonality between such device variants is a consensus on the internal physical properties. In general, the randomness observed in resistive switching devices can be predominantly attributed to the probabilistic generation and rupture of conductive filaments (CFs) and underlying physical elements [26,27,28]. As illustrated in Figure 2a, under the influence of an externally applied voltage, the metal undergoes oxidation at the anode, which causes the migration of metal ions to form particle clusters, followed by reduction at the cathode to create CFs. The CF model is currently the most frequently and widely studied. The internal physical mechanism will be elaborated in detail in the subsequent section. Based on the filament’s composition, it can be broadly classified into metal-based and non-metal-based filaments. In the former, metals typically serve as the conductive medium, whereas in the latter, defects such as oxygen vacancies are generally utilized as conductive particles. Due to the kinetic behavior of diffusion, CFs are generated and ruptured stochastically, giving rise to devices with oscillatory electrical behavior. Specifically, the stochastic characteristics of the CFs are primarily manifested in different distributions of conductance and threshold voltage, which approximately conform to a normal distribution (Figure 2b,c). This feature endows these devices with unique advantages in certain practical applications.

The random entropy source in the device arises from inherent probability behavior, leading to high levels of randomness, non-repeatability, and unpredictability. This feature enhances the security of built-in hardware security primitives, making them comparable to biometric data in terms of their robustness and reliability. Liu et al. [29] demonstrated a Bi_2_O_2_Se-based memristor, in which threshold voltages exhibited highly random variation, and this randomness was harnessed to construct true random number generators (TRNGs) for security applications. Ding et al. [30] proposed a unified approach to combine TRNGs and physical unclonable functions (PUFs) in a 4-layer 3D NbO_x_ memristor array. This approach is robust against machine learning-based attacks. Furthermore, biological microcircuits are inherently stochastic, and neuroscience research showed that injecting noise into learning and information processing can lead to efficiency gains. [16] The ability of memristors to simulate neuronal noise and stochastic neuronal dynamics at the device level holds great promise for the creation of dense neural populations that can robustly represent signals and neuronal states. This will facilitate the development of neuromorphic computers with highly integrated memory and processing units. [31] Importantly, an ideal mathematical and physical model is needed to model the memristor-based intrinsic oscillatory behavior for theoretical verification of more complex algorithms and hardware platforms, thereby promoting the development of efficient and secure neuromorphic computing.

### 2.2. Conductive Filament Mechanism

The primary challenge in establishing the intrinsic variation model of the memristor lies in elucidating its inherent physical mechanism. Despite the existence of experimental data, the complete comprehension of the resistive switching (RS) mechanism remains ambiguous at this stage. Based on research findings, the current classification of memristor models can be categorized into three mechanisms: the CF mechanism [32,33,34], the interface type switching [35], and the electronic effect mechanism [36,37,38,39]. Among the three memristor mechanisms, the CF mechanism involves the random generation of CFs within the dielectric layer, where random variability is a prerequisite for realizing CS. Due to its extensive research, the CF model is the main focus of this section. Within the CF mechanism, the RS process stems from the growth and rupture of CFs in the switch layer, as illustrated in Figure 2a. Based on the type of CF, three RS mechanisms are introduced, namely the electrochemical metallization mechanism (ECM) [40,41], the valence change mechanism (VCM) [42,43], and the phase change mechanism (PCM) [44,45], as depicted in Figure 3.

#### 2.2.1. Electrochemical Metallization Mechanism (ECM)

Typical ECM cells, also known as conductive bridging random access memory (CBRAM) [25], are composed of metal–insulator–metal layer stacks. In ECM, the active electrode is generally located at the top, while the inert electrode is located at the bottom. The switching between them is achieved through the electrochemical growth and dissolution of small metal channels between the inert and active electrodes. As illustrated in Figure 3a, the filament growth process for Ag cation migration and reduction is depicted. The filament originates from the inert electrode (top) and, during the subsequent growth process, a limited number of cations will preferentially diffuse to the existing filament end due to the highest electric field in front of the protrusion. This process, in turn, results in branched growth with dendrites pointing towards the active electrode, which is defined by twisted electric field lines (bottom). Figure 3b,c displays a transmission electron micrograph (TEM) image. Figure 3b illustrates the CFs formation process from a bottom electrode (BE) to a top electrode (TE). Upon connecting TE and BE, the resistance shifts from a low resistance state (LRS) to a high resistance state (HRS) (Figure 3d). Conversely, applying a negative voltage to TE results in the Ag atoms within the CF dissolving into Ag^+^ via an oxidation reaction, rupturing the CF (Figure 3c), and abruptly decreasing the current (Figure 3e), leading to a switch from LRS to HRS. Depending on the positions of electrons in Ag^+^ and BE during the reduction reaction, the growth kinetics determine whether the Ag CF grows from TE or BE. When the mobility of Ag^+^ surpasses that of electrons, Ag^+^ and electrons will be situated near BE, resulting in Ag CF growth from BE to TE [46,47]. Moreover, CFs can break spontaneously due to their instability, with some exceptions. Song et al. [48] proposed an Ag/TiO_2_-based memristor and observed volatile electrical behavior, which can be explained by the spatial repulsion between the silver filament and the surrounding oxide lattice, resulting in the fragility of CFs. Upon decreasing or withdrawing the voltage, the rapid contraction of the elongated Ag nanoparticle cluster or joule heating when the voltage is swept back to zero can naturally break CFs on the bottom electrode. Additional explanations for ECM-based volatile behavior include Rayleigh instability, minimization of surface energy, and the Gibbs–Thomson effect caused by the surface diffusion of metal atoms [46].

#### 2.2.2. Valence Change Mechanism (VCM)

The VCM mechanism also relies on the formation and fracture of CFs for the conductance transition, which is similar to the ECM mechanism. In VCM, the switching layer is composed of oxide, and the electrodes typically use inert electrodes or oxides (Figure 3f). Under the influence of an electric field, oxygen vacancies migrate and organize to form non-metallic CFs, resulting in a switch in the conduction state. Using ZnO as an example, the growth process of CFs was observed via in situ TEM. When an external voltage is applied, the electrochemical reaction generates doped oxygen ions (O^2−^) that move toward the Pt anode electrode. Moreover, as the sample is subjected to a higher voltage, O^2−^ is repelled and oxygen vacancies accumulate at the cathode (top electrode, TE). Once a sufficient concentration of oxygen vacancies accumulates near the electrode and the charge is balanced by biased electrons, the oxygen vacancies rearrange to form an ordered structure and grow towards the bottom electrode [49]. The entire process is illustrated in Figure 3g. In this case, the RS behavior is due to the migration of oxygen ions, which results in a switch between ZnO_1−x_ and ZnO dominated by Zn. Additionally, the redistribution of oxygen vacancies tunes the stoichiometry of the switching layer of the oxide, thereby regulating the conductivity of the film. VCM devices usually have more stable CFs, and better erasing and writing capabilities because the metal electrode is inert and hardly participates in ion migration, and no impurities remain in the oxide layer [43].

#### 2.2.3. Phase-Change Mechanism (PCM)

Phase-change materials are highly effective storage materials that can be mass-produced [50]. Phase-change memristors consist of a small volume of a phase-change material that is sandwiched between two electrodes. These materials exhibit a unique switching behavior between a crystalline state, characterized by low resistivity, and an amorphous state, characterized by high resistivity, which is triggered by applying an electrical pulse of sufficient magnitude to generate the necessary heat for the phase transition. To produce amorphous regions within the crystalline matrix, a reset pulse of sufficient amplitude (referred to as a reset pulse) is applied such that the current flowing through the device generates Joule heating, which melts most of the phase-change material [32]. If the pulse is abruptly terminated, the molten material quickly quenches into an amorphous phase due to the glass transition. Chalcogenide glass, namely GST (Ge_2_Sb_2_Te_5_), is a widely used material for phase change memory and is used as an example in this section [51]. Figure 3h displays the cross-sectional TEM of a phase-change memristor in the full set, partially reset, and fully reset states. Upon the application of the reset pulse, the device transitions from LRS to the HRS, causing an increase in the volume of the amorphous region at the top of the bottom electrode, until the bottom electrode is completely covered and the device state changes to HRS. The inset displays the diffraction pattern of the amorphous region, indicating that GST is polycrystalline. Following the reset pulse, the device state is dominated by the high resistance of the amorphous GST region, and the electron diffraction pattern in the fully reset state is depicted in Figure 3i. Moreover, in certain special materials such as VO_2_, the device automatically reverts to a HRS after the set pulse is removed or attenuated; this phenomenon can be explained by the Peierls-type phase transition [52].

Clearly, in the CF mechanism, the electrical properties of the memristors are directly determined by CFs. As the formation of CFs is inherently stochastic, memristors based on CFs often exhibit intrinsic variability, which needs to be accounted for in physical models for theoretical validation of CS.

### 2.3. Physical Model for Stochasticity Distributions

The intrinsic oscillatory behavior of memristors is mainly manifested through probabilistic switching and random conductance distribution. Numerous studies reported normal distribution of conductance states and threshold voltages, which were applied in information processing and security [53,54,55,56,57,58,59]. The high level of randomness in the conductance states and/or threshold voltages of memristors was utilized to construct random encoding matrices for information encoding and computation, including compression and encryption. Therefore, it is essential to establish a reliable model that can theoretically justify the use of information encoding and computation. As the CF mechanism is currently the predominant model, this section focuses on the concept of relying on a single or dominant CF structure to trigger the resistance transition mechanism.

Taking typical titanium oxide memristors as an example [60], Figure 4a depicts a single cell of Pt/TiO_2_/Pt structure memristors and its conduction mechanism. Upon the application of a positive bias, a redox reaction takes place causing Ti^4+^ to transform into Ti^3+^ near the anode. The Ti^3+^ ions, in the form of Ti_4_O_5_^2+^, then drove towards the other electrode and reacted with O^2−^ ions, forming Ti_2_O_3_, the metastable phase of titanium dioxide. Consequently, Ti_2_O_3_ accumulates at the cathode and forms high CFs that grow towards the anode. In this type of memristor, the conductance G is determined by the length of the CFs, with longer growth lengths resulting in reduced overall drag. The reverse reaction occurs when a negative voltage is applied. Based on the switching mechanism, the conductance G can be expressed as a function of the two titanium materials’ structure. Qian et al. [61] based on this mechanism established a related mathematical model:(1)$$G=\frac{1}{R}=\frac{A}{\rho_{1}l+\rho_{2}(d-l)}$$
where *R* is the total resistance of the memristor, and *A* and *d* represent the cross-sectional area and thickness of the Ti nanowire, respectively, $\rho_{1}$ and $\rho_{2}$ are the resistivities of the CFs and the high-resistivity TiO_2_, respectively, while *l* represents the length of the grown filament. *A* and *d* usually have large variations due to the variations in line edge roughness and thickness fluctuation from the fabrication processes and are typically modeled as Gaussian distributions. In addition, the filament growth can be expressed as:(2)$$f_{i}=\frac{t_{switch}}{\Delta t},\;l=\sum_{x=0}^{fi}a_{x}$$
where *f_i_* is the number of growth iterations under an applied pulse of duration *t_switch_*, and Δ*t* is the single iteration time, which is a constant determined by the intrinsic properties of titanium. The filament length *l* is accumulated for each growth iteration, denoted by *a_x_*. Since write operations can be short, usually *t_switch_* is not uniform. In addition, *a_x_* is also affected by *q*, *m**, *ω*, *τ*, *V_0_,* and *κ* factors, as shown in the following analysis model [15].
(3)$$a_{x}=\frac{qV_{0}}{2m^{*}}\times [\begin{array}{c} \frac{1}{d\;-\;(a_{1}\;+\;a_{2}\;+\;\cdots \;+\;a_{x-1})}(sin\omega t_{x-1}-sin\omega t_{x})\\ \frac{\kappa }{1+{\left(\omega \tau \right)}^{2}}\frac{1}{d}[(sin\omega t_{x-1}-\omega \tau cos\omega t_{x-1})\\ -(sin\omega t_{x}-\omega \tau cos\omega t_{x})] \end{array}]\times {\left(\Delta t\right)}^{2}$$
(4)$$t_{x}=x\times ∆t$$
where *q* and *m** are the electron charge and its effective mass, both of which should be regarded as constants; *ω* and *τ* are the frequency and mean free time between two consecutive collisions, which are determined by the intrinsic properties of titanium; *V*_0_ is the applied voltage; *κ* is given by the Arrhenius equation, *κ = Ae^−Eac/RT^*, which can introduce temperature variations between different memristors during the initialization phase. Figure 4b displays the conductance distribution based on the model, where the conductance values exhibit a lognormal distribution. The conductance values present a lognormal distribution. The selected parameter configurations are closely aligned with the actual scenario in device fabrication. Additionally, models exist that were proposed to validate the log-normal distribution of conductance [27,62]. The emergence of normal distribution behavior is crucial for CS implementation, as elaborated in Section 4.

Furthermore, as discussed in the previous section on mechanism analysis, the CF model is applicable to both volatile and nonvolatile memristors. Therefore, the electronic behaviors of both types of memristors follow a normal distribution, which is supported by numerous experimental data [63,64,65]. For instance, Huang et al. [66] utilized a kinetic Monte Carlo simulation to model the LRS of Ag/Ta_2_O_5_:Ag/Pt and Ag/Ta_2_O_5_/Pt memristors, as illustrated in Figure 4c,d, respectively. The simulation was based on ion movement and redox reactions that form CFs. The resistance distribution of the insulating Ta_2_O_5_ and conductive Ag points were solved using the continuity equation, and the potential of each point was calculated to establish the probability of ion movement or redox reaction. The simulated conductance distributions of the LRS for the Ag/Ta_2_O_5_:Ag/Pt and Ag/Ta_2_O_5_/Pt devices presented normal distributions, as depicted in Figure 4e. Figure 4c illustrates that the Ag nanoclusters and filaments in the Ag/Ta_2_O_5_:Ag/Pt device can be considered bipolar electrodes when an electric field is applied. Due to the acceleration of the redox reaction by the bipolar electrode, it is easier to alter the morphology of the filament and the distribution of Ag nanoclusters in the Ag/Ta_2_O_5_:Ag/Pt device, leading to increased randomness of the LRS conductance, which is advantageous for the realization of CS.

## 3. Memristor Crossbar Arrays for Matrix-Vector Multiplication

In a large memristor crossbar array, memristors can leverage the laws of physics to perform MVM in a single step, making them particularly suitable for applications in neural networks [67,68,69,70], matrix equation solving [71,72], and signal encoding [15,73,74,75]. This capability is also crucial for implementing CS algorithms in memristor arrays. Figure 5a illustrates the memristor crossbar array, where the input vector and corresponding matrix can be mapped to input voltages and the conductance of the memristor array, respectively. By utilizing Ohm’s law and Kirchhoff’s current law for multiplication and accumulation operations, respectively, the MVM results can be obtained in a single step, leading to a significant improvement in calculation speed and energy consumption, as shown in Figure 5b. The matrix computing capability of memristor crossbar arrays can be utilized to solve matrix equations and achieve significant acceleration. Zidan et al. [71] conducted experimental studies to solve Poisson’s equation using memristor crossbar arrays (as depicted in Figure 6a). The solution obtained after 10 iterations demonstrated a perfect match with the expected solution, as presented in Figure 6b,c. Sun et al. [72] illustrated that a cross-point memristor array can perform the matrix-inversion operation to solve a system of linear equations. Figure 6d shows the matrix-inversion operator circuit, and the comparison of the measured output voltage of this circuit with the analytical solution x = A^−1^b reveals an error below 10% (Figure 6e). Wang et al. [76] proposed a scalable massively parallel computing scheme that exploits continuous-time data representation and frequency multiplexing in nanoscale interleaved arrays. This approach enables a one-time parallel read of stored data and MVM in interleaved arrays (as shown in Figure 7a). The errors of parallel reading for this system are presented in Figure 7b,c. The error is less than 2% at a small voltage amplitude and is comparable to that reported in neuromorphic computing (Figure 7b). Furthermore, the error decreases with increasing frequency, and the signal-to-noise ratio increases with frequency (Figure 7c). This system, when combined with signal modulation performed concurrently with parallel signal processing, can be used for low-power smart edge applications. In conclusion, memristors can provide scalable parallel computing systems with high processing speed, power efficiency, and low computational errors. This approach is particularly effective in achieving CS acceleration and addressing the problem of explosive growth in the number of sensors from the root.

## 4. Memristor Arrays for Compressed Sensing

### 4.1. Compressed Sensing

Donoho proposed a groundbreaking sampling compression technology called CS [77]. CS is an architecture that integrates compression and perception and is considered one of the important frontier technologies in the development of artificial intelligence. The traditional Nyquist theorem employs equidistant uniform sampling, which requires the sampling frequency to be twice the lowest frequency of the signal to prevent signal aliasing. This greatly limits the sampling speed [78]. The CS algorithm uses random non-uniform sampling and does not require the signal to be extended at a fixed period in the frequency domain. This enables the signal to be compressed while sampling, thus breaking through the limitations of Nyquist’s theorem and enabling high-rate sampling [4,79]. Compared to the traditional full sampling recompression process, low power consumption and high-efficiency data processing can be achieved. The special structure of the memristor array enables the realization of a scalable parallel computing system that can perform MVM in a single step. Therefore, it can be used to realize a parallel CS system [80,81,82]. The foundation of the parallel CS algorithm is as follows.

When the processing object is a two-dimensional signal such as an image, the signal can be compressed and reconstructed in each column of the image through a measurement matrix [83]. Assume that the original signal *X* is a $\sqrt{N}$ × $\sqrt{N}$ real number, which can be regarded as composed of *N* one-dimensional vectors in *R_N × 1_*. Assume that x_i_ represents the *i_th_* column of *X*, and Φ of *M × N* dimensions represents the measurement matrix (Figure 8a). Then, the sub-sampling process of parallel CS can be expressed as:(5)$$y_{i}=\Phi x_{i},i=1,\;\ldots ,\;N$$
where *y*_i_ is the measurement value corresponding to each column of *X*, the dimension is *M* × 1, and the combination of all column measurement values *y_i_* is the entire measurement value *Y*; that is, *Y* = [*y*_1_, *y*_2_, …, *y_N_*]. For CS, there is a premise that the signal approximately satisfies sparsity; or as long as the signal satisfies approximate sparsity on a certain transform domain, compression can be achieved in the sparse domain. Since actual natural images are rarely absolutely sparse, it is often necessary to perform a sparse transformation on natural images to complete CS in its sparse domain. Ψ is based on a sparse orthogonal matrix of size *N × N*. Any signal in an R_N_ can be represented by the basis of an *N* × 1 vector. Any signal can be expressed as:
(6)$$x_{i}=\Psi s_{i},i=1,\;\ldots ,\;N$$
where $s_{i}$ is the vector of the weighting coefficients, and the entire weight *S* is the concatenation of all $s_{i}$. *X* and *S* exactly represented equivalent signals. *X* and *S* are in the time domain and Ψ domain, respectively (Figure 8a). Based on the discussion above, the specific CS process of each column can be expressed as:
(7)$$y_{i}=\Phi x_{i}=\Phi \Psi s_{i}=\Theta s_{i}$$
where Θ is the sensing matrix with θ=ΦΨ (Figure 8b). Moreover, the most critical condition is that Θ must meet the restricted equidistant properties (RIP) rules; otherwise, the reconstruction of the compressed signal cannot be completed [84]. The RIP rules are as follows:
(8)$$(1-\delta_{k}){∥x∥}_{2}^{2}\leq {∥\Theta x∥}_{2}^{2}\leq (1+\delta_{k}){∥x∥}_{2}^{2}$$
where the equidistance is constant $\delta_{k}$ ∈ (0, 1), *k* is the number of coefficients. The essence of the RIP rule is that the measurement matrix and the sparse matrix are uncorrelated. Generally speaking, the Gaussian matrix and Bernoulli distribution matrix are simple and easy to implement and can satisfy the RIP rule with a high probability [85]. Unfortunately, there is currently no algorithm to verify whether the RIP rules are met, and it can only be verified through simulation. Moreover, $y_{i}$ is the compression vector with the size of *M* and $x_{i}$ is the original vector with the size of *N*. Because of *M < N*, the equation is indeterminate. Therefore, this is an optimization problem solved with the *l1*-norm for evaluating the signal.(9)$$\hat{s}=argmin{∥s^{\prime }∥}_{1}\;\mathrm{such}\;\mathrm{that}\;\Theta s^{\prime }=y$$

In the end, it is necessary to transform the sparse signal into a time-domain signal *X*. The sampling and reconstruction processes of parallel CS are processed column by column, which significantly reduces the size of the array and computational complexity.

Currently, numerous studies demonstrated that memristor-based CS hardware exhibits higher efficiency and lower power consumption than traditional complementary metal-oxide semiconductors (CMOS). Wang et al. [86] demonstrated that the RRAM-based interleave structure enables 10 times faster speed, 17 times higher energy efficiency, and three orders of magnitude smaller area compared to CMOS. Memristor-based CS hardware implementations can be broadly categorized into volatile memristors and non-volatile memristors.

### 4.2. Non-Volatile Memristor for Compressed Sensing

The implementation of CS with non-volatile memristors relies on programming the conductance values in the crossbar array, which mainly involves Bernoulli distribution matrices and Gaussian distribution matrices. The programmed conductance matrix G serves as the measurement matrix Φ. The sensory signal is then converted into an electrical signal and fed into the memristor array, thereby completing the sub-sampling function. The sensory signal can be compressed near the sensor, reducing the cost of data transmission while ensuring information security (which will be discussed in the next section) [87]. As discussed above, implementing CS requires the measurement matrix to follow either Gaussian or Bernoulli distribution, and memristors can realize the switching function with conductance represented by Gaussian distribution. Hence, memristors are excellent candidates for implementing CS. Two methods are available for programming the memristor measurement matrix: the first method involves randomly programming the memristors on and off in the array to realize the Bernoulli distribution; the second method involves setting all the memristors to LRS and using Gaussian conductance distribution to deploy the measurement matrix. Recent research demonstrated that memristor arrays can perform not only compression but also on-chip signal reconstruction. Gallo et al. [88] proposed a novel method to implement fast and robust compression and reconstruction with approximate message passing using in-memory computing. Figure 9a illustrates the memristor crossbar array for the compression process. The compression measurement process involves applying x_0_ as a voltage to the array to complete the MVM row through digital-to-analog conversion, superimposing the output current obtained on the column, and obtaining y through analog-to-digital conversion. The on-chip reconfiguration process is shown in Figure 9b. The approximate message passing (AMP) algorithm runs on a dedicated processing unit, while the matrix equation solving is deployed on the memristor array. To fully demonstrate the feasibility of the memristive CS system, Qian et al. [89] proposed a framework that successfully integrates the memristive CS system with the CMOS image sensor system, as shown in Figure 9c. This architecture greatly reduces the amount of data from the image sensor before digitization and wireless transmission, thus saving power and accelerating performance. However, it should be noted that the memristor-based CS system requires the integration of two arrays to form a bipolar structure to implement the positive and negative elements in the measurement matrix, thus achieving low power consumption and transmission cost. Furthermore, the current CS system can only use a fixed sampling rate or an offline algorithm to modify the calculation rate. Since sensing images have varying weights and sparsity, a fixed compression ratio may reduce efficiency and reconstruction quality. Qian et al. [89] proposed a compression rate (CR) self-adaptive design to enable CR to be self-adaptive at runtime, as shown in Figure 9d. The CS adaptive encoding for the 1920 × 1080 image sensor is achieved by judging whether there is a significant difference between the input segment and the previous input segment to determine whether a larger data sampling matrix is needed. The significant progress made in non-volatile memristors inspired researchers to explore more devices, structures, and algorithms.

### 4.3. Volatile Memristor for Compressed Sensing

Recent research highlighted the potential of volatile memristors in enabling tunable random sampling strategies and cost-effective measurement implementations. This is due to their ability to spontaneously return to HRS, which can be leveraged to construct CS systems. Volatile memristors have a random switching behavior and a Gaussian conductance distribution, which makes them suitable for CS applications. There are currently two approaches to implementing CS using volatile memristors: constructing a stochastic sampling circuit or creating a measurement matrix based on the Gaussian conductance distribution in HRS. A key requirement for volatile memristors to be utilized in CS is high durability, as they need to function continuously. Bao et al. [90] developed a tunable stochastic oscillator using a TiN/VO_2_/TaO_x_/Pt volatile memristor, which can be used as a core control device for CS. The I-V curve of the VO_2_/TaO_x_ stacking structure and the sampling circuit are illustrated in Figure 10a,b, respectively. When the memristor is turned on and the voltage exceeds EN, the transmission gate (TG) is turned on to complete the signal sampling. The recovery performance of the tunable stochastic oscillator (TSO)-based CS paradigm is compared to the traditional Nyquist method in Figure 10c,d. The results demonstrate that TSO has a higher compression ratio and lower error rate in implementing a CS scheme. Furthermore, Wang et al. [91] developed a polyimide memristor to realize CS. The conductance of the device exhibits a Gaussian distribution when returning from an LRS to HRS, which can be used to construct a conductance matrix as a measurement matrix in CS. The schematic circuit is illustrated in Figure 10e. In this approach, image pixels are first mapped to an input voltage of 0~0.2 V, which is then input row by row into the crossbar array to complete the compression process, followed by reconstruction at the receiving end. Moreover, this work also demonstrated that the one-time sampling (OTS) matrix, constructed using the intrinsic oscillatory behavior of the device, can make the compressed signal satisfy perfect security in information theory. This implies that encryption can be embedded in the compression layer at no additional cost. The details of this will be discussed in the following section.

The non-volatile memristor enables the implementation of CS with a single programming step, and deployment can be quickly achieved using Ohm’s and Kirchhoff’s laws. However, the repeated use of the same measurement matrix may result in the complete exposure of compressed information to unauthorized users, thereby reducing signal security. In contrast, volatile memristors enable tunable random sampling strategies and cost-effective measurement implementations by spontaneously returning to HRS, which can enhance the security of CS systems. However, volatile memristors require repeated programming of the measurement matrix before each sampling, resulting in higher power consumption in low safety requirement scenarios. Therefore, users can choose different memristive-based CS hardware based on their specific needs, whether they prioritize safety or energy efficiency.

## 5. All-in-One Memristor-Based Compression and Cryptosystem

A separate encryption layer can be too costly due to the increasing number of sensors and the limited resources of each sensor. To address this issue, integrating data protection mechanisms into the information awareness stage to achieve simultaneous compression and encryption is an ideal solution [92]. CS exploits the structure of a specific signal and encodes it by utilizing a random sub-sampling operator on the physical interface between the analog and digital domains. The complete signal cannot be recovered if the random sampling operator is kept secret. To accurately recover the signal, the receiver must know the actual encoding matrix Φ used in the random sub-sampling encoding when decoding the CS measurements. Therefore, if the Φ matrix is only shared with the receiver through a private channel, it can be used as a key to simultaneously compress and encrypt the signal. Unauthorized users cannot obtain the correct sparsity by solving a convex optimization problem without knowledge of Φ [87]. Recently, studies showed that changing the measurement matrix can provide additional security against chosen-plaintext attacks (CPA) and achieve perfect security [93,94]. Although changing the measurement matrix can resist CPA and achieve perfect security, programming the memristor conductance can be costly. Wang et al. [91] proposed a solution in which the conductance of volatile memristors changes each time it returns to the HRS. Importantly, it can spontaneously return to the HRS, enabling the implementation of OTS pad encryption for perfect security. Compared to non-volatile memory, this approach does not require a secondary programming process, significantly reducing the cost of encryption. The article reports that for 50% compression of a 512 × 512 image, the programming voltage required to ensure security is 0.392 W for volatile memristors, while non-volatile memristors require double that amount. However, without OTS encryption, the cost of implementing CS is only 80 nW.

Furthermore, to expedite the CS process, parallel CS is commonly employed, whereby CS is conducted on the signal line by line. However, with this approach, the features of each column are preserved in that specific column, even though the OTS matrix guarantees perfect security. As illustrated in the red box in Figure 11a, this can result in the exposure of edge features in simple images, thereby rendering them non-confidential. To address this issue, Figure 11b proposes an integrated scheme for OTS compression and encryption based on volatile memristors. A highly random key is generated through the random threshold-switching behavior of memristors. This approach requires only one diffusion operation on the compressed data, which not only ensures absolute security and can withstand CPA attacks, but also effectively prevents information leakage [93,95,96]. The effectiveness of the diffusion behavior can be observed in Figure 11a, where the edge information is completely obscured. This presents a promising solution for applications that demand high efficiency and security.

## 6. Conclusions and Prospect

Memristor-based CS systems for edge computing are still in the early stages of development and face both opportunities and challenges. To construct a more efficient memristor-based edge computing paradigm, researchers are exploring various schemes for designing devices, arrays, and systems that can implement CS.

From a device perspective, the probability of RS and Gaussian conductance are essential factors for implementing CS. Relevant models based on the CF mechanism are analyzed and discussed to demonstrate the feasibility of the memristor-based CS system in a scientific manner. From an array perspective, the MVM operation can be implemented on memristor-based crossbar arrays in the analog domain, with low power consumption, low complexity, and high speed. Additionally, the inherent variability of memristors allows measurement matrices to be easily embedded in the arrays, and MVM operation can be performed with attractive advantages, ultimately accelerating the implementation of CS. Compared to traditional CMOS, the memristor-based CS system exhibits faster speed and lower power consumption. In a memristor-based CS system, for scenarios with low-security requirements, the non-volatile memristor only requires a single programming step to obtain a fixed measurement matrix for the implementation of the CS system. On the other hand, the volatile memristor requires programming before each sampling to generate the OTS measurement matrix. As the measurement matrix changes with each sampling, this can enhance system security and is suitable for use in scenarios with high-security requirements. Subsequently, CS can effectively achieve simultaneous compression and encryption by incorporating a data protection layer into the information awareness stage. Based on this technique, the ciphertext can resist CPA and satisfy perfect security with the OTS matrix. Moreover, the OTS matrix can be spontaneously implemented using volatile switching behaviors, significantly reducing the cost of encryption without requiring a secondary programming process. Additionally, a simple exclusive-OR (XOR) diffusion can prevent parallel CS from the risk of leaking plaintext edge information, providing an excellent solution for applications requiring high efficiency and safety.

In the future, memristor-based crossbar arrays offer advantages in various areas of signal processing, such as signal filtering, time-frequency transformation, and information classification. When combined with CS, it is expected to realize an efficient edge computing architecture that integrates compression, encryption, storage, and recognition. Therefore, the new computing paradigm based on memristors is quite attractive for edge-computing systems. However, there are still several challenges that need to be addressed to fully exploit the potential of memristor-based arrays for CS. Firstly, improvements in memristor devices can be made by increasing their switching speed, reducing the operating voltage and power consumption, improving device yield, and suppressing the off-state current. Moreover, the physical and mathematical model of the multi-layer structure devices should be established to further optimize the measurement matrix and guide the deployment of the CS algorithm. At the crossbar array level, the IR-drop and latent paths pose significant obstacles to the application of large memristor arrays. Therefore, it is essential to research integration schemes with transistors and selectors to prevent misreading and reduce IR-drop, which is critical for the deployment of measurement matrices. Moreover, the impact of IR-drop on the quality of image reconstruction needs to be further evaluated to establish a suitable IR-drop interval that allows the deployment of large arrays of CS hardware. At the algorithm and system level, the original image contains redundant data, and techniques such as image enhancement, segmentation, and feature extraction are utilized to obtain image prior information, quickly eliminate image background noise, and focus on sampling feature areas to improve efficiency. Additionally, the potential path of the device and the IR-drop defects may degrade the quality of image reconstruction; hence, it is necessary to design a reconstruction algorithm with more stability, lower computational complexity, and fewer observation times to accurately restore the compressible signal with noise in real-world complex environments. Finally, even though volatile memristors can simultaneously perform CS and XOR diffusion encryption to achieve perfect security and resist CPA, these two operations are still separate. Thus, a new integrated scheme is required to enable both CS and diffusion for the efficient and secure deployment of an edge computing paradigm.

## Figures and Tables

**Figure 1 nanomaterials-13-01354-f001:**
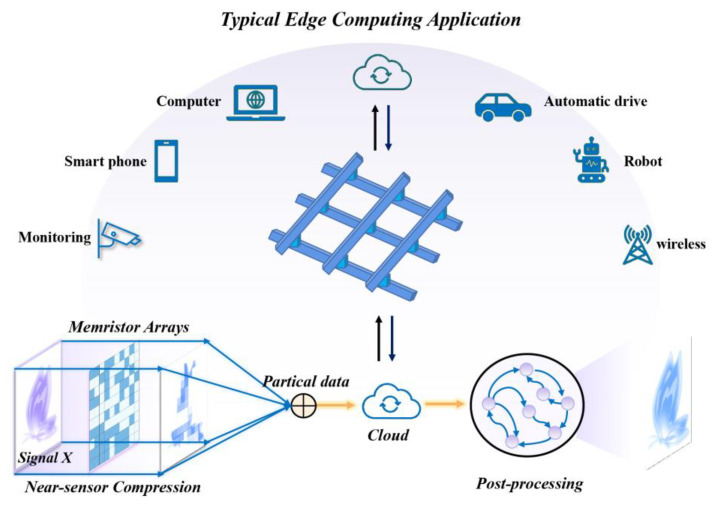
Illustration of the compressed sensing based on memristor crossbar arrays for typical edge computing application. Reproduced with permission. [2] Copyright 2022, Wiley−VCH.

**Figure 2 nanomaterials-13-01354-f002:**
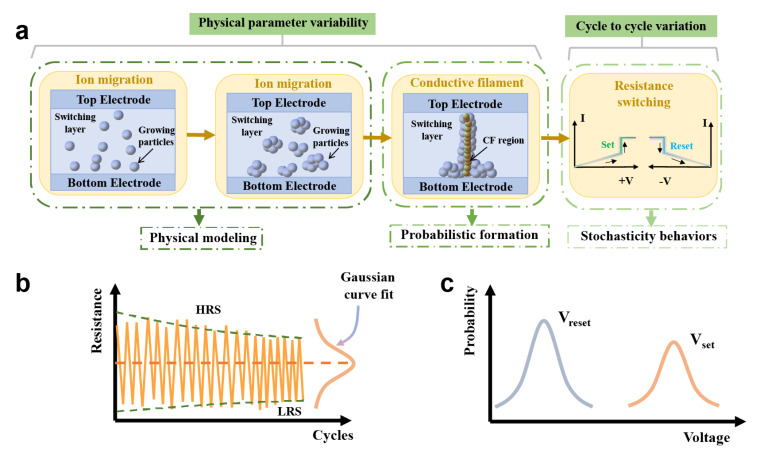
Resistive switching characteristics and oscillating electrical behaviors. (**a**) The memristor resistive switching model includes four stages: ion migration, accumulation, conduction channel formation, and resistance switching. The propagation of ions along the entire device size is random, and the growth of conductive filaments is also generated probabilistically, resulting in an oscillating behavior of resistive switching characteristics. (**b**,**c**) This inherent variation in the probabilistic formation and rupture of CF results in an oscillatory behavior of conductance (**b**) and threshold voltage (**c**) across cycles, which can be represented by a Gaussian distribution.

**Figure 3 nanomaterials-13-01354-f003:**
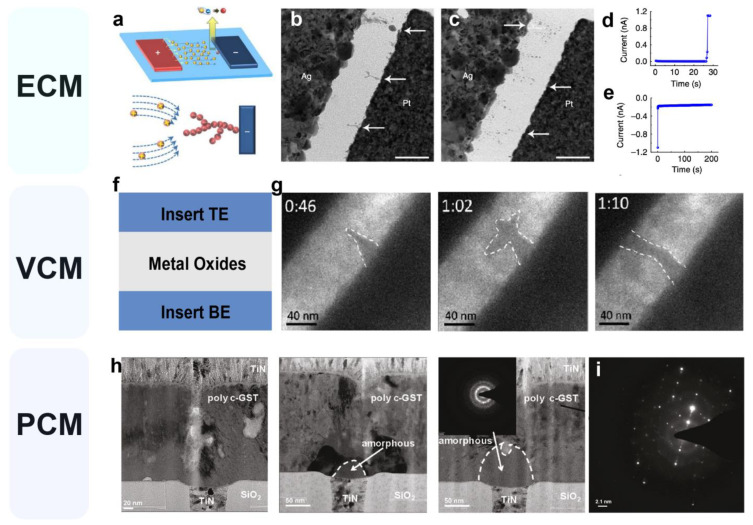
Memristor resistance switching mechanism with three different types. (**a**) Schematic of dendritic CF formation in the Ag/SiO_2_/Pt structure. (**b**,**c**) TEM image of Ag/SiO_2_/Pt device after set (**b**) and reset (**c**) process. Arrows illustrate CF formation. (**d**,**e**) Correspondence of I-t curve to CF annihilation process. Reproduced with permission [47] Copyright 2012, Springer Nature. (**f**) VCM device structure diagram. (**g**) In situ TEM image of filament growth in ZnO thin film VCM device. Reproduced with permission [49] Copyright 2013, American Chemical Society. (**h**) Cross-sectional TEM image of a GST-based electronic synapse. (**i**) The electron diffraction pattern in the fully reset state. Reproduced with permission [51] Copyright 2013, American Chemical Society.

**Figure 4 nanomaterials-13-01354-f004:**
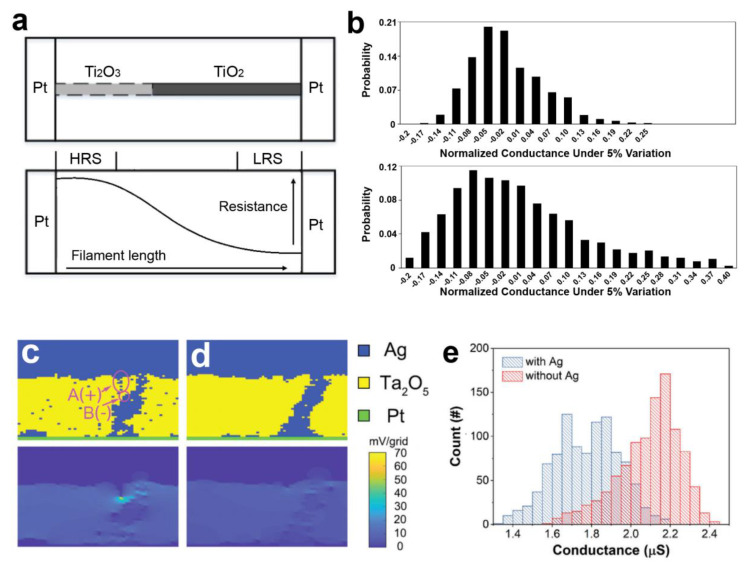
The memristor model with conductive filament mechanism. (**a**) A generic Pt/TiO_2_/Pt memristor device model. The top is the conductive filament growth model and the bottom is the illustration of filament length and device resistance. (**b**) The simulation device conductance conforms to the Gaussian distribution for the model. Reproduced with permission [61]. (**c**,**d**) Simulation of random origins observed in Ag/Ta_2_O_5_:Ag/Pt and Ag/Ta_2_O_5_/Pt devices. (**e**) Simulated conductance distribution for the LRS of the Ag/Ta_2_O_5_:Ag/Pt and the Ag/Ta_2_O_5_/Pt devices. Reproduced with permission. [66] Copyright 2020, Wiley−VCH.

**Figure 5 nanomaterials-13-01354-f005:**
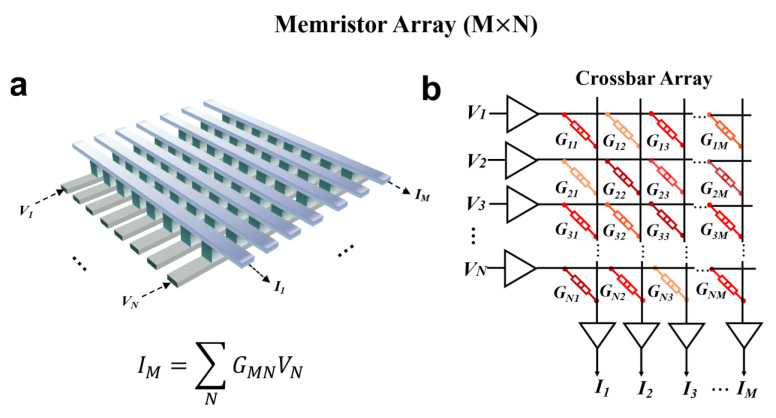
The memristor arrays of M × N for matrix-vector multiplication. Schematic of a memristor crossbar (**a**) arrays and (**b**) circuit. The voltage sequences (1 × N) are applied to the arrays to multiply by the conductance matrix G (N × M) to achieve MVM operation. According to Ohm’s law and Kirchhoff’s equations, the element I (1 × M) is output in one step by listing.

**Figure 6 nanomaterials-13-01354-f006:**
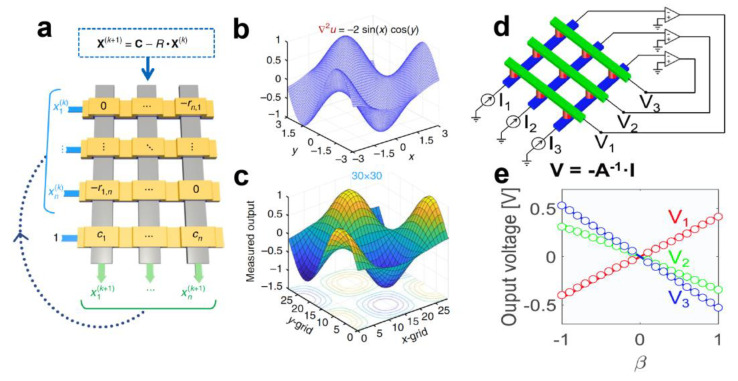
Memristor arrays for solving matrix-vector equations. (**a**) Memristor crossbar-based system for solving a Poisson’s equation. (**b**,**c**) The solution after 10 iterations shows an excellent match with the expected solution. Reproduced with permission [71] Copyright 2018, Springer Nature. (**d**) Memristor cross-point circuit for solving matrix equations in one step. Circuits to calculate a scalar product I = G·V by Ohm’s law, and to calculate a scalar division *V = −I/G* by a TIA. (**e**) The parameter *β* to control the input current given by *I* = *β* [0.2; 1; 1] *I_0_* with −1 ≤ *β* ≤ 1. Reproduced with permission [72] Copyright 2019, National Academy of Sciences.

**Figure 7 nanomaterials-13-01354-f007:**
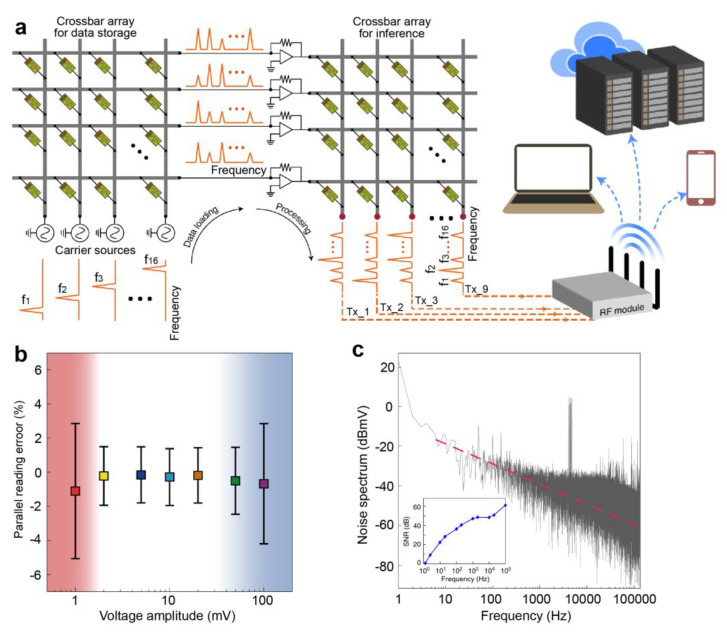
Memristor arrays for frequency multiplexing computing technology. (**a**) Frequency multiplexing computing technology based on two memristor crossbar arrays for numerous data storage and inference. (**b**) Relative error of the carrier signal for different voltage amplitudes. (**c**) The noise suppression with increasing operating frequency. Reproduced with permission [76] Copyright 2021, Springer Nature.

**Figure 8 nanomaterials-13-01354-f008:**
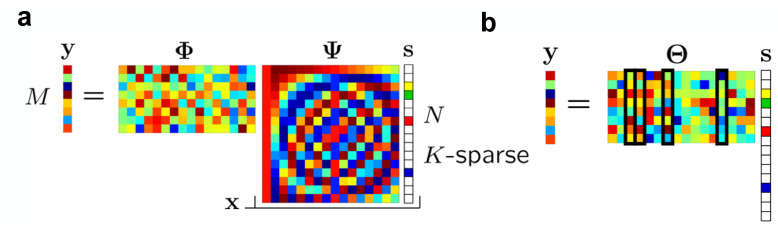
The compressed sensing processes. (**a**) Compressed sensing measurement process with measurement matrix Φ and discrete cosine transform (DCT) matrix Ψ. (**b**) The matrix product Θ=ΦΨ with the four columns corresponding to nonzero *S_i_* highlighted. Reproduced with permission [4] Copyright 2007, IEEE.

**Figure 9 nanomaterials-13-01354-f009:**
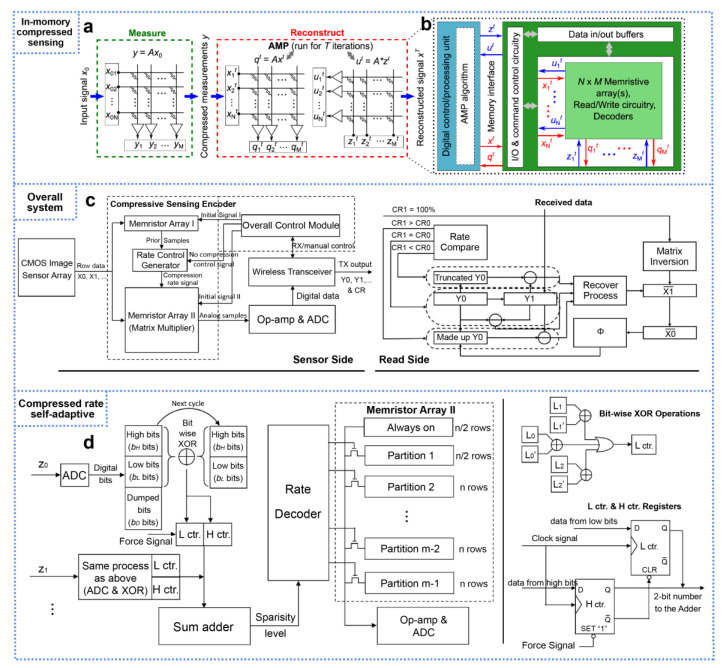
Non-volatile memristors for compressed sensing. (**a**) The N × M memristive crossbar encodes the measurement matrix A, which is used to obtain the CS measurements and implement the MVM of the AMP recovery algorithm. (**b**) Architecture of the memristive implementation of AMP. Reproduced with permission [88] (**c**) The overall architecture based on the memristor arrays for the CS encoder. Reproduced with permission. Reproduced with permission. (**d**) The CR self-adaptive system to make CR self-adaptive at runtime. Reproduced with permission. [89] Copyright 2020, ACM.

**Figure 10 nanomaterials-13-01354-f010:**
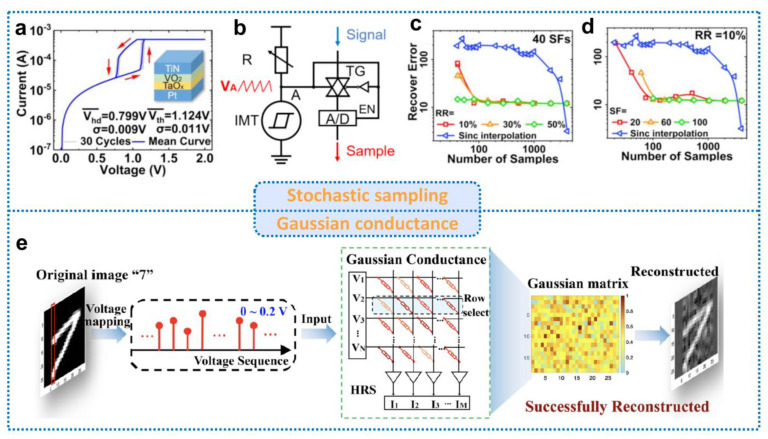
Volatile memristors for compressed sensing. (**a**) The typical I-V curve of VO_2_/TaO_x_ stacking structure. (**b**) Schematic image of the stochastic sampling circuit. (**c**,**d**) Benchmark of TSO-based compressed sensing paradigm with conventional Nyquist method. Reproduced with permission [90] Copyright 2020, IEEE. (**e**) Gaussian conductance mode for signal compression. The original image “7” was mapped as a 0–0.2 V voltage sequence and applied to an array of devices with a Gaussian conductance distribution. Signal compression coding is accomplished by Kirchhoff’s current law and Ohm’s law. Finally, the OMP algorithm reconstructs the signal. Reproduced with permission [91] Copyright 2023, Wiley−VCH.

**Figure 11 nanomaterials-13-01354-f011:**
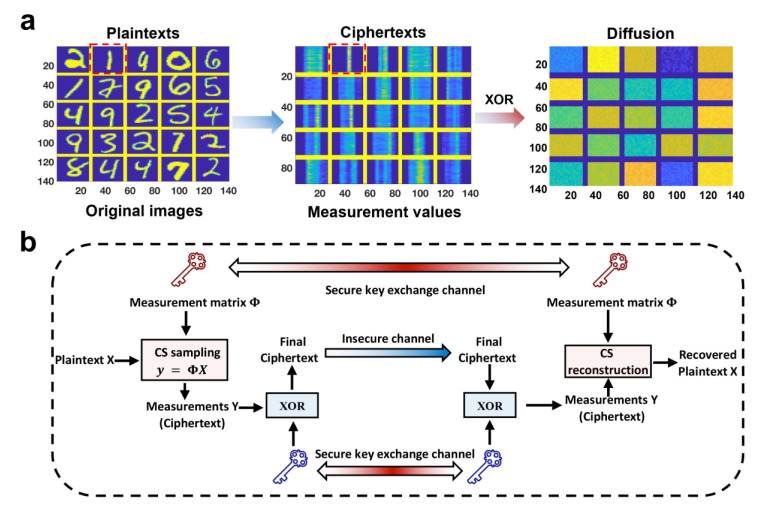
An XOR diffusion operation to implement a secure all-in-one compression and encryption system. (**a**) The original signals (plaintext), the measurements (ciphertext), and diffusion result by XOR. The measurements after CS expose edge features and a diffusion operation makes the edge information hidden. (**b**) Schematic diagram of a perfectly secure all-in-one compression and encryption engine system based on CS. Reproduced with permission [91] Copyright 2023, Wiley−VCH.

## Data Availability

All data are contained within this article.

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
