# Peer review of "Memristor-Based Signal Processing for Compressed Sensing"

_nanomaterials, 2023, doi:10.3390/nano13081354_

Round 1

Reviewer 1 Report

The idea to exploit the intrinsic properties of memristors to develop advanced computing abilities in a compressed sensing (CS) approach is very recent but incredibly attractive. Although not so many papers are in the literature, the idea to realize a review is worth it due to the increasing interest and potentialities. The Authors initially introduce the general mainframe, then explain the memristor main physical working principles and how these can match the CS approach requirements. CS is briefly presented, with main memristor types and architectures that can fulfill CS requirements, finally giving the basis for an intriguing all-in-one compression/encryption system. Strengths and weaknesses are clearly presented, with the conclusions suggesting other applications of the proposed approach. The paper quality is high, the references are updated, and the scientific messages are clear. Just a few issues must be addressed before publication, in more detail:

1.       Line 468: 0.392W is the power consumption.

2.       Always define acronyms, e.g., MVM, TE, RIP.

Reviewer 2 Report

The manuscript is a collection of artificially associated elements and concepts, without consistent arguments. It should be cleaned by extrapolations and structured more coherently.

Reviewer 3 Report

The authors presented a comprehensive overview of the use of memristive devices for the application of compressed sensing. The device requirement for CS and the physical mechanism of memristive devices are well elaborated. The authors also demonstrated the summary of non-volatile and volatile devices in the application of CS. The manuscript concludes with the summary of the memristive devices and future research directions for signal processing, which is very instructive for the researchers in the related fields. The reviewer has no further comment on this manuscript as it is well organized and clearly described. 

Round 2

Reviewer 2 Report

..